# Investigation of the Accuracy of Four Intraoral Scanners in Mandibular Full-Arch Digital Implant Impression: A Comparative In Vitro Study

**DOI:** 10.3390/ijerph19084719

**Published:** 2022-04-13

**Authors:** Adolfo Di Fiore, Lorenzo Graiff, Gianpaolo Savio, Stefano Granata, Michele Basilicata, Patrizio Bollero, Roberto Meneghello

**Affiliations:** 1Department of Neurosciences, Section of Prosthetic and Digital Dentistry, University of Padova, 35100 Padova, Italy; lorenzo.graiff@unipd.it (L.G.); stefano.granata@unipd.it (S.G.); 2Departments of Civil, Environmental and Architectural Engineering, University of Padova, 35100 Padova, Italy; gianpaolo.savio@unipd.it; 3Department of Systems Medicine, University of Rome Tor Vergata, 00133 Roma, Italy; michele.basilicata@ptvonline.it (M.B.); patrizio.bollero@ptvonline.it (P.B.); 4Department of Management and Engineering, University of Padova, 35100 Padova, Italy; roberto.meneghello@unipd.it

**Keywords:** CAD/CAM, digital impression, intra-oral scanner, accuracy, full arch, dental implant

## Abstract

Background: We compare the accuracy of new intraoral scanners (IOSs) in full-arch digital implant impressions. Methods: A master model with six scan bodies was milled in poly(methyl methacrylate), measured by using a coordinate measuring machine, and scanned 15 times with four IOSs: PrimeScan, Medit i500, Vatech EZ scan, and iTero. The software was developed to identify the position points on each scan body. The 3D position and distance analysis were performed. Results: The average and ± standard deviation of the 3D position analysis was 29 μm ± 6 μm for PrimeScan, 39 μm ± 6 μm for iTero, 48 μm ± 18 μm for Mediti500, and 118 μm ± 24 μm for Vatech EZ scan (*p* < 0.05). Conclusions: All IOSs are able to make a digital complete implant impression in vitro according to the average misfit value reported in literature (150 μm); however, the 3D distance analysis showed that only the Primescan and iTero presented negligible systematic error sources.

## 1. Introduction

The main factor for the survival and success of an implant-supported fixed dental prosthesis (FDP) is the obtainment of a passive fit between the prosthesis and the dental implant [1]. It is difficult to give an exact numerical value to the misfit due to the different findings present in the literature. The value published has a range from 10 μm [2] to 150 μm [3]; however, the misfit should be around 30–50 μm to avoid mechanical and biological complications [4].

Several factors may influence the passive fit; however, the impression procedure represents the first critical steps [5,6]. The traditional impressions with different techniques are considered the gold standard in implant-supported FDPs [7]. The introduction of digital technologies in dentistry has changed daily clinical practice. An increase in accuracy and patient comfort and decreased operative time and clinical treatment are the main characteristics of the digital technologies described in the literature [8,9,10]. However, doubts remain among clinicians regarding the accuracy of the intraoral scanners (IOSs) used to scan a full arch for implant-supported fixed dental prostheses. Several authors have demonstrated the trueness and precision of the digital impressions realized by using different IOSs in a full arch [4,11,12,13,14]; however, it is difficult to compare the results due to the different assessment methodologies and master models. In recent years, new IOSs have been introduced to and others removed from the dental market. Digital scanning has been reported to be challenging, particularly in the multiple implants scenario [15,16]. Therefore, the need to know the performance of different IOSs is crucial for clinicians to obtain an accurate digital impression and consequently a passive fit in the full-arch implant-supported FDPs [16]. The purpose of the present study was to compare different intraoral scanners on the accuracy of mandibular full-arch digital implant impressions.

## 2. Materials and Methods

The authors used the same master model and assessment procedures to allow the comparison among IOSs, according to previous research conducted [4]. The master model was designed using a computer-aided design (CAD) software (SolidWorks, Dassault Systèmes SolidWorks Corporation, Waltham, MA, USA). Six scan bodies were incorporated into the master model, positioned vertically and symmetrically at different heights (Figure 1).

The geometry of all scan bodies was a cylinder with a diameter of 4 mm. All the central axes were parallel to each other. The favorable design was chosen to easily and accurately perform the calibration measurements using a coordinate measuring machine (CMM). Moreover, the shape geometry of the scan bodies stressed the stitching algorithm/procedure adopted by scanning systems [17,18]. After the design phase, the master model was milled in poly(methyl methacrylate) using a 5-axes milling machine (Dyamach Italia s.r.l., Mussolente, VI, Italy). Silicone (Vestogum, 3M Espe, St. Paul, MN, USA) was used to simulate the soft tissue. The methodology consisted of three steps.

First, the master model was measured 5 times using a CMM (OGP SmartScope Flash CNC 300, Rochester, NY, USA) with the contact system through a probe with a ruby sphere of 1.5 mm in diameter [6,16,17]. The scan-body was measured on the upper and lateral surfaces to place the 3-dimensional (3D) coordinate in the digital environment [4,17,18]. The coordinates of the probed points were transferred into a 3D CAD geometric modeling software program (Rhinoceros 5.0, Robert McNeel & Associates, Seattle, WA, USA) and analyzed to estimate the position and orientation of each scan body through a plug-in realized in IronPython by using the upper plane and central axis of the cylinder of each scan body [4]. The perpendicular intersection of the geometrics figures generated a point that represented the position point of each scan body. The average position point of each scan body achieved with the CMM was used as a reference point to compare the digital impressions obtained by using the 4 IOSs [4]. In the second step, the master model was scanned with four new different IOSs: iTero Element (Align Technology Inc., San Jose, CA, USA, software version 5.7.0.301), Medit i500 (Medit corporation, Seoul, South Korea, software version 2.0.3), Vatech EZ scan (Tecno-Gaz spa, Parma, Italy, software version 2.0.1.29), and PrimeScan (Sirona Dental System GmbH, Bensheim, Germany, software version 5.1). The scanning was performed according to the manufacturer’s instructions for each scanner. Fifteen digital impressions were made.

The third step consisted in postprocessing and analysis of the data. The digital impressions in the standard triangulation language (STL) format were sent to Geomagic Studio Software (Geomagic Studio 12, 3D Systems Inc., Rock Hill, SC, USA) to clean the mesh from acquisitions not related to the research. The files were imported in the plug-in called “Scan-abut” of the CAD geometric modeling software (Rhinoceros 6.0, Robert McNeel & Associates, Seattle, WA, USA) to perform the 3D position and 3D distance analysis [4]. The “Scan-abut” software identified the position point on each scan body using the tool to intersect the upper plane and central axis of the cylindric. Automatically the software aligned the position points of the scan bodies of the digital impression and of the master model to calculate the 3D deviation among them. The analysis was performed for each digital impression realized with the 4 IOSs [4].

The distance analysis consisted of calculating the 3D intra scan bodies distance, using the position points, among paired scan bodies (i.e., distance from scan abutment 1 to scan abutment 6). A total of fifteen 3D distances, considering any combinations of six scan bodies, were calculated for each digital impression. The difference between the 3D distance between scan bodies of the digital impression and the reference 3D distance between scan bodies of the master model was considered as the 3D distance error [4].

The accuracy was assessed according to the normative ISO 5725-1 [19] and -2 [20]. The mean deviation (error) of the 3D position was considered as the trueness, while the standard deviation of the 3D position errors relevant to the group sample (i.e., fifteen digital impressions) was the precision. The distance error was used to evaluate the relationship between the error and the distance between the abutments as an indicator of the maximum permissible error (MPE) of the scanning system in accordance with ISO 10360 standards [4,17,18]. The digital impression was considered as the statistical unit. The primary variable was the 3D position error (μm). Six numerical values were recorded for each impression, which corresponded to the deviations of the six scan bodies; then, for each impression, the average of the six-position errors was calculated to obtain a single numerical value. The 3D mean position error was used in comparative statistics. The Wilcoxon matched-pairs signed-rank test (one-tailed) was used to compare IOSs. The level of statistical significance was set as α = 0.05, with a statistical power of 80%. The 3D distance error was used in regression analysis. Statistical analysis was performed using statistical software SPSS 16.0 (SPSS Inc.; Chicago, IL, USA).

## 3. Results

The descriptive statistic of the 3D position errors of each IOS is given in Table 1.

The mean 3D position error values were used in comparative statistics between digital impressions. Statistically, a significant difference emerged between all the IOSs (*p*-value < 0.05). The 3D distance analyses of different IOSs are reported in Figure 2. The 3D distance analysis showed a linear relationship between distance errors and scan-body distance only with the devices PrimeScan and iTero.

## 4. Discussion

Intraoral scanners have become everyday equipment in the dentistry world; therefore, the accuracy of the digital impression is an important factor for the success and survival of an implant-retained prosthesis [1,2,3,4,5,6,7,8,9].

The 3D position analysis showed that all IOS were valid for executing digital impressions for a full arch, according to the clinically desirable value of the position errors in a full-arch rehabilitation reported in the literature [2,3,4]. PrimeScan, iTero Element, and Medit i500 had very good performance, while the Vatech EZ scan had an acceptable performance, according to the average value of misfit published in literature [2,3,4].

The 3D distance analysis allowed individuation of the presence of systematic and/or random errors. Primescan and iTero showed regression lines close and almost parallel to the x-axis, which meant that the systematic errors sources were negligible. Instead, the other IOSs (Vatech EZ scan and Medit i500) presented a strong linear relationship between error and scan-body distance due to both systematic and random influencing factors. The random errors were not identifiable. Hardware, software, environment, algorithms, stitching, and file processing are the main factors that might generate random errors. Moreover, the scan strategy could influence the accuracy during the digital impression, [21,22] such as the scan-body geometry [14].

Clinically, the IOS should be considered as a measurement system with its relevant measurement errors, which cannot be completely removed. The clinician–operator should prefer to have reduced random errors and fully compensated systematic errors or systematic errors, which may be identified and successively corrected in the design or production phase of the prosthetic. Therefore, when the error is systematic and known, we can set the workflow to obtain an accurate dental prosthesis. Instead, it is not possible to control the accuracy when there are significant random errors in the impressions. Though the 3D position analysis reported that all IOSs were valid for executing digital impressions for a full arch, according to the clinically desirable value of the position errors reported in the literature, the 3D distance analysis showed the difference between the IOSs. The 3D distance analysis should therefore be considered as an in-depth investigation of the 3D position analysis to understand the possibility of further improving the performance of the scanning system. Therefore, the evaluation of the accuracy of new devices using the 3D position analysis should be considered as the first analysis step.

To the best of our knowledge, this article assessed four IOSs presented in the dental market. The evaluation method processing was performed automatically and, thus, independently of the operator. The object of the authors is to provide clinicians some indications of the new devices’ ability to acquire a full arch. PrimeScan presented the best performance regarding the 3D position errors compared to the other IOS. The 3D position accuracy for the IOSs in a full-arch implant-supported fixed dental prosthesis was as follows: PrimeScan (26 ± 6 μm), iTero Element (39 ± 6 μm), Medit i500 (48 ± 18 μm), and Vatech EZ scan (118 ± 24 μm). The same authors, in a previous article [4], assessed the accuracy of other IOSs. The authors reported that not all IOSs can be used for digital impressions in full-arch implant-supported fixed dental prostheses. TrueDefinition (31 ± 8 μm) and TRIOS3 (32 ± 5 μm) presented better performance regarding the 3D position analysis. However, the 3D distance analysis showed a good relationship between error and scan-body distance only with the TrueDefinition. Other authors have investigated the accuracy of IOSs in a full-arch digital impression. Renne et al. [23] used seven different scanners and concluded that the order of trueness for complete arch scanning was as follows: 3Shape D800 (43.6 μm) > iTero (56.2 μm) > 3Shape TRIOS3(69.4 μm) > Carestream 3500 (76 μm) > Planscan Plameca (96.2 μm) > CEREC Omnicam (101.5 μm) > CEREC Bluecam (140.5 μm). Imburgia et al. [11] concluded that the CS 3600 had the best trueness (60.6 ± 11.7 μm), followed by Cerec Omnicam (66.4 ± 3.9 μm), TRIOS 3 (67.2 ± 6.9 μm), and True Definition (106.4 ± 23.1 μm). Bilmenoglu et al. [13] tested 10 IOSs and concluded that only some of the digital scanners had the necessary performance for the fabrication of complete-arch implant-supported fixed dental prostheses. Only the group formed by Omnicam, TRIOS devices, Apollo DI, and Bluecam scanners presented an accuracy within the range of 31 to 45 μm. However, the previous articles did not all investigate the same IOS. Mangano et al. [12], in recent research, found the same results as our study. The following IOS: iTero, PrimeScan, CS 3700, CS3600, and TRIOS 3 had an average deviation of <40 μm and represent a theoretically compatible solution for taking impressions for full-arch restorations. However, it is difficult to compare our results with other studies due to different analysis methodologies.

A limitation of this study was that all scanners were used to scan the same model. To interpret the present study results, it should be considered that saliva, blood, the movement of the mandibula, tongue, and the patient may complicate the quality of a digital scan [11,21]. However, further clinical verifications are needed to understand the accuracy of these devices and to confirm the performance of these IOSs.

## 5. Conclusions

Based on the findings of this in vitro study, the following conclusions were drawn:The 3D position analysis showed that all IOSs were able to execute digital impressions for a full arch, according to the clinically desirable value of the position errors reported in the literature (150 μm).The 3D distance analysis showed that the Primescan and iTero presented regression lines close and almost parallel to the x-axis, which meant that the systematic errors sources were negligible.

## Figures and Tables

**Figure 1 ijerph-19-04719-f001:**
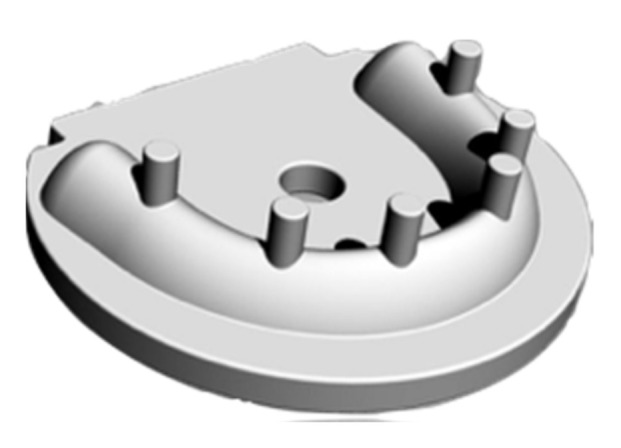
The design of the master model in a digital environment.

**Figure 2 ijerph-19-04719-f002:**
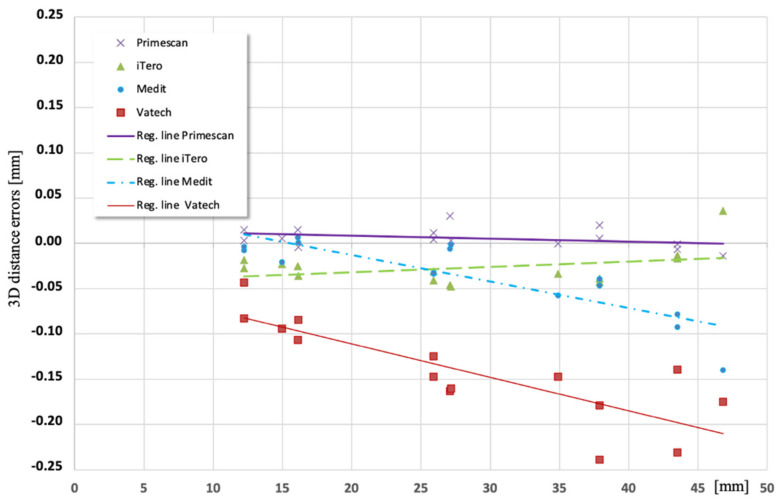
3D distance errors: distance errors [mm] vs. inter-abutment distance [mm] with regression line of errors for each scanner.

**Table 1 ijerph-19-04719-t001:** 3D position errors of the digital impressions. * difference statically significance (*p*-value < 0.05).

IOS	∆X Mean (SD) [μm]	∆Y Mean (SD) [μm]	∆Z Mean (SD) [μm]	3D Mean (SD) [μm]
PrimeScan	4.6 (3.4)	10.4 (7.8)	1.4 (0.9)	29 (6) *
iTero Element	15.4 (3.81)	28.1 (18.3)	4.9 (2.1)	39 (6) *
Medit i500	36.3 (27.41)	14. 4 (7.5)	4.2 (3.5)	48 (18) *
Vatech EZ Scan	69.5 (27.1)	60 (41.1)	22.7 (6.4)	118 (24) *

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
