# Peer review of "Investigation of the Accuracy of Four Intraoral Scanners in Mandibular Full-Arch Digital Implant Impression: A Comparative In Vitro Study"

_ijerph, 2022, doi:10.3390/ijerph19084719_

Round 1
Reviewer 1 Report
The manuscript presents accuracy of newly 4 intraoral scanners in 2 mandibular full-arch digital implant impression. I am surprised why I agreed to review this MS. While the manuscript is well written and having originality, there are some issues that need to be addressed. The most surprising aspect of the manuscript is that the MS is of just 4 pages excluding the bibliography part with one figure and one table only, so I doubt whether the manuscript belongs to full length article category or a short communication. The results need to be elaborated more. The whole discussion part has been repeated twice which editor should have checked before sending it for review. The discussion is very confusing and does not explain the hypothesis with respect to the results of the MS and previous findings. English language used is fair, however, it is the responsibility of the authors to make sure that the manuscript should possess similarity index which should be acceptable by the editor as per policies of the journal to avoid any plagiarism related issue that could creep in knowingly or un-knowingly. The authors are suggested to confirm this by employing any reliable plagiarism check program/software.
Author Response
Thank you for your suggestions and observations. The article is the second step of research on the accuracy of several intraoral scanners ( Di Fiore, A.; Meneghello, R.; Graiff, L.; Savio, G.; Vigolo, P.; Monaco, C.; Stellini, E. Full arch digital scanning systems performances for implant-supported fixed dental prostheses: a comparative study of 8 intraoral scanners. J Prosthodont Res 2019,63,396-403), therefore, this manuscript may be considered a short communication. The discussion has been modified in the revised manuscript and I attach the result of the plagiarism check made

Reviewer 2 Report
The manuscript describes the results of four current intraoral scanners for application on full-arch implant-supported prosthesis. It is an important topic to update the accuracy of new equipments.
Some major points are listed below:
Lathan et al. (2020) and Favero et al. (2019) demonstrated that the scanning technique influences the accuracy on a dentate arches. Since it may occur when scanning scanbodies, it should be important to refer the scanning technique used in the present study.
Mizumoto et al. demonstrated in 2020 that the scanbody geometry influence the accuracy of intraoral scanning. This topic should be included in the discussion.
Author Response
Thank you so much for your suggestions and observations. The
The following references have been entered and discussed in the manuscript.
Latham J, Ludlow M, Mennito A, Kelly A, Evans Z, Renne W. Effect of scan pattern on complete-arch scans with 4 digital scanners. J Prosthet Dent. 2020 Jan;123(1):85-95. Favero R, Volpato A, Francesco M, Fiore AD, Guazzo R, Favero L. Accuracy of 3D digital modeling of dental arches. Dental Press J Orthod. 2019 Jan-Feb;24(1):38e1-37e7. /Mizumoto RM, Yilmaz B, McGlumphy EA Jr, Seidt J, Johnston WM. Accuracy of different digital scanning techniques and scan bodies for complete-arch implant-supported prostheses. J Prosthet Dent. 2020 Jan;123(1):96-104
Reviewer 3 Report
The authors present an in vitro study describing the in vitro evaluation of four intraoral scanners. The authors used a master model, and the accuracy of the scanners was evaluated.
The English language needs correction throughout the manuscript. See, for instance, lines 33, 58 and 80.
Lines 23-25: I do not agree with this sentence, as I find it may confuse the readers. Since this is an in vitro study, the statement suggesting all the IOS are valid for complete implant impression can extrapolate the obtained results.
The authors report that although one author found misfit values of 150μm acceptable, most authors reported lower values (around 30-50μm) as acceptable. Since one of the scans (Vatech) reported values of 118±24μm, which is considerably higher than these values, do the authors think it can be clinical acceptable?
Line 51: when referring to newly what timeframe is being considered?
Since so many different IOS are available what was the rationale to select this four?
I suggest the materials and methods being illustrated with images from the master model, the 3D geometric modelling with the central axis of each cylinder and the imposition between the control and the digital impressions.
Table 1: please add the meaning of IOS in the table caption. Also, I suggest adding the observed statistical differences, with * or other symbols.
Pages 4 and 5 present the same figure and repeated text. It was clearly an error.
Lines 211-212: I do not agree with this sentence. Since this is an in vitro study, the statement suggesting all the IOS are valid for complete implant impression can extrapolate the obtained results.
Author Response
Thank you so much for your suggestions and observations. In the following sentences, we try to answer your questions.
Q: The English language needs correction throughout the manuscript. See, for instance, lines 33, 58 and 80.
A: The sentences have been corrected and re-written
Q: Lines 23-25: I do not agree with this sentence, as I find it may confuse the readers. Since this is an in vitro study, the statement suggesting all the IOS are valid for complete implant impression can extrapolate the obtained results.
A: The sentence has been changed with the following “ All IOSs are able to make a digital complete implant impression in vitro, however, the 3D distance analysis showed that only the Primescan and iTero presented systematic errors sources negligible.”
Q:The authors report that although one author found misfit values of 150μm acceptable, most authors reported lower values (around 30-50μm) as acceptable. Since one of the scans (Vatech) reported values of 118±24μm, which is considerably higher than these values, do the authors think it can be clinical acceptable?
A: The sentence has been changed with the following “All IOSs are able to make a digital complete implant impression in vitro according to the average misfit value reported in literature (150 mm)”
Q: Line 51: when referring to newly what timeframe is being considered?
A: The word “newly” has been deleted.
Q: Since so many different IOS are available what was the rationale to select this four?
A: The 4 IOSs selected for this research represent the last devices do not analyze in the previous article published “ Di Fiore, A.; Meneghello, R.; Graiff, L.; Savio, G.; Vigolo, P.; Monaco, C.; Stellini, E. Full arch digital scanning systems performances for implant-supported fixed dental prostheses: a comparative study of 8 intraoral scanners. J Prosthodont Res 2019,63,396-403.”
Q:I suggest the materials and methods being illustrated with images from the master model, the 3D geometric modelling with the central axis of each cylinder and the imposition between the control and the digital impressions.
A:Thank you for your suggestion, however, the images regarding the procedure has been reported in the previous article published “Di Fiore, A.; Meneghello, R.; Graiff, L.; Savio, G.; Vigolo, P.; Monaco, C.; Stellini, E. Full arch digital scanning systems performances for implant-supported fixed dental prostheses: a comparative study of 8 intraoral scanners. J Prosthodont Res 2019,63,396-403.” In the manuscript the following sentence has been entered “ The authors used the same master model and the assessment procedures to allow the comparison among Ios according to previous research conducted[4].”
Q:Table 1: please add the meaning of IOS in the table caption. Also, I suggest adding the observed statistical differences, with * or other symbols.
A: The table 1 has been corrected.
Q_Pages 4 and 5 present the same figure and repeated text. It was clearly an error.
A: The picture has been deleted.
Q:Lines 211-212: I do not agree with this sentence. Since this is an in vitro study, the statement suggesting all the IOS are valid for complete implant impression can extrapolate the obtained results
A:The conclusion has been changed with the following sentences: “ Based on the findings of this in vitro study, the following conclusions were drawn: The 3D position analysis showed that all IOSs are able for executing digital impressions for a full arch according to the clinically desirable value of the position errors reported in the literature (150 mm).
Round 2
Reviewer 1 Report
The authors have addressed the queries raised. However, I will go with the decision of the editor.